# Metabolomics-guided discovery of cytochrome P450s involved in pseudotropine-dependent biosynthesis of modified tropane alkaloids

Radin Sadre [1,2], Thilani M. Anthony [2], Josh M. Grabar [1], Matthew A. Bedewitz[1,3], A. Daniel Jones [2] & Cornelius S. Barry [1✉]

Plant alkaloids constitute an important class of bioactive chemicals with applications in medicine and agriculture. However, the knowledge gap of the diversity and biosynthesis of phytoalkaloids prevents systematic advances in biotechnology for engineered production of these high-value compounds. In particular, the identification of cytochrome P450s driving the structural diversity of phytoalkaloids has remained challenging. Here, we use a combination of reverse genetics with discovery metabolomics and multivariate statistical analysis followed by *in planta* transient assays to investigate alkaloid diversity and functionally characterize two candidate cytochrome P450s genes from *Atropa belladonna* without a priori knowledge of their functions or information regarding the identities of key pathway intermediates. This approach uncovered a largely unexplored root localized alkaloid sub-network that relies on pseudotropine as precursor. The two cytochrome P450s catalyze *N*-demethylation and ring-hydroxylation reactions within the early steps in the biosynthesis of diverse *N*-demethylated modified tropane alkaloids.

[1] Department of Horticulture, Michigan State University, East Lansing, MI 48824, USA. [2] Department of Biochemistry and Molecular Biology, Michigan State University, East Lansing, MI 48824, USA. [3]Present address: Department of Chemical and Biological Engineering, University of Colorado, Boulder, CO 80303, USA. ✉email: barrycs@msu.edu

Plants have evolved dynamic metabolic pathways producing a plethora of structurally and functionally diverse small specialized metabolites to cope with biotic and abiotic stresses. Some species that are high producers of these phytochemicals have become an integral part of human civilizations as sources for drugs, nutraceuticals, flavors, fragrances, and cosmetics. Select genera of the Solanaceae (Nightshade) family represent invaluable sources of biologically active tropane alkaloids, including the clinically important anticholinergics and antispasmodics atropine (racemic hyoscyamine) and scopolamine[1]. These basic nitrogen-containing compounds constitute a specific class of specialized metabolites sharing an N-methylated bicyclic tropane core structure (8-methyl 8-azabicyclo[3.2.1]octane). Several species of the Solanaceae family, including the major crop plants tomato, potato, eggplant, and pepper, accumulate modified tropane alkaloids named calystegines[2–4]. These are potent glycosidase inhibitors with potential applications in the treatment of metabolic diseases[5,6]. Unlike hyoscyamine and scopolamine that are esters formed from N-methylated tropine (3α-tropanol) and phenyllactic acid, calystegines are polyhydroxylated, N-demethylated nortropane alkaloids with an 8-azabicyclo[3.2.1]octane core derived from pseudotropine (3β-tropanol)[1]. The sequence of reaction steps yielding calystegines is, however, unknown.

Advances in genomics and transcriptomics facilitate the identification of candidate genes in plant alkaloid biosynthesis, which has paved the way for biotechnological production of some plant alkaloids in alternative hosts[7–10]. However, many phytoalkaloid biosynthetic pathways have not been sufficiently investigated to allow for biotechnological optimization or biosynthetic pathway reconstruction. Major challenges lie in the functional characterization of candidate enzymes because alkaloid production often proceeds via largely unknown multi-step pathways in plants, and key enzymes' substrates have yet to be discovered. In addition, alkaloid biosynthetic pathways usually involve enzymes from the large cytochrome P450 superfamily that catalyze a wide range of chemical modifications. Eukaryotic cytochrome P450s are typically membrane-bound enzymes located at the endoplasmic reticulum that depend on co-localized cytochrome P450 reductases for electron transfer from NADPH and whose activity is modulated by the membrane and cytosolic environment[11–13]. These features often hinder the production of functional plant cytochrome P450s in conventional microbial hosts and as a consequence, only about 200 functionally distinct plant cytochrome P450 genes have been identified to date, of which about one tenth are involved in alkaloid biosynthesis[14].

In the past decade, the medicinal plant Atropa belladonna (Deadly Nightshade) emerged as a model system to investigate tropane alkaloid biosynthesis in the Solanaceae. Tropine-dependent biosynthesis of hyoscyamine/scopolamine occurs in the roots of A. belladonna, and mining of a publicly available A. belladonna transcriptome dataset for genes with root-preferential expression pattern discovered seven of the thirteen structural genes involved in putrescine and phenylalanine dependent hyoscyamine/scopolamine production[9,15–18]. In fact, all known structural genes for hyoscyamine/scopolamine biosynthesis are either exclusively or preferentially expressed in secondary roots of A. belladonna. The knowledge of the sequence of reaction steps and enzyme catalysts recently led to successful reconstruction of the biosynthetic pathway in yeast[9]. Alkaloid diversity is under-investigated in plants, and enzyme catalysts for the production of modified tropane alkaloids with new or altered bioactivity and pharmacological properties remain to be discovered. We previously identified A. belladonna tropinone synthase (AbCYP82M3), a cytochrome P450 that catalyzes the formation of the bicyclic ring system of tropinone (8-methyl-8-azabicyclo[3.2.1]octan-3-one), the precursor of tropine and pseudotropine[16]. The study initially considered the root-preferentially expressed cytochrome P450 genes encoding AbP450-5021 and AbP450-116623 as candidates, but further analyses revealed both cytochrome P450s lack tropinone synthase activity (Supplementary Fig. 1)[16]. In the present study, we combine reverse genetics with discovery metabolite profiling and multivariate statistical analysis to investigate potential functions of two candidate A. belladonna cytochrome P450 genes, encoding AbP450-5021 (CYP71 family) and AbP450-116623 (CYP82 family), in alkaloid biosynthesis in planta. These two candidate genes were selected solely based on their root-preferential expression. The metabolomics-based approach discovered modified tropane alkaloids in a largely unexplored alkaloid sub-network associated with modifications of pseudotropine cores and revealed distinct roles for AbP450-5021 and AbP450-116623 in this sub-network. We used Nicotiana benthamiana as a platform for in planta transient assays to verify the activities of AbP450-5021 and AbP450-116623 and explore reactions with candidate substrates. This platform has several important advantages: N. benthamiana has a system in place for electron transfer to the non-native, functionally active plant cytochrome P450s and can be engineered to co-produce target enzymes for pathway re-construction. A. belladonna and N. benthamiana are both members of the Solanaceae but from different sub-families, and non-engineered N. benthamiana does not produce tropane alkaloids or modified tropane alkaloids[16,19,20]. The analyses revealed catalytic transformations of pseudotropine-derived intermediates: AbP450-5021 has a dual role as an N-demethylase/ring-hydroxylase while AbP450-116623 is a ring-hydroxylase. In A. belladonna root, these cytochrome P450s catalyze distinct, early steps in the pseudotropine-dependent biosynthesis of hydroxylated and/or esterified N-demethylated nortropane alkaloids. The findings demonstrate that the metabolomics-guided approach taken here leads to functional characterization of candidate genes/enzymes when there is limited prior knowledge of complex biosynthetic pathways and intermediate identities. Our strategy has the potential to accelerate the discovery of genes and engineering of underlying biosynthetic pathways for biotechnological production of high-value compounds.

## Results

**Discovery of modified tropane alkaloids in *A. belladonna* root.** To investigate the roles of *AbP450-5021* and *AbP450-116623*, we developed virus-induced gene silenced (VIGS) *A. belladonna* lines in which the expression of each gene was suppressed separately (Supplementary Figs. 2 and 3). We next examined root tissue from VIGS and control lines for metabolite composition through untargeted ultra-high performance liquid chromatography/high-resolution mass spectrometry (UHPLC/HRMS). The data were processed for peak detection and integration, chromatographic alignment, data reduction and normalization, and signals for compounds were subjected to multivariate statistical analyses. To distinguish metabolites whose levels were altered by silencing, orthogonal projections to latent structures-discriminant analysis (OPLS-DA) models were built from the *AbP450-5021* VIGS/control lines and *AbP450-116623* VIGS/control lines data sets. In these models, VIGS and control lines clustered in two well-separated groups according to their genotype (Fig. 1a, b). Representative OPLS-DA S-plots were generated to recognize root metabolites that contributed to the discrimination of the lines (Fig. 1c, d). Feature ions were subsequently filtered using the p(corr) values obtained from OPLS-DA models, retaining signals with p(corr) values below −0.3 and greater than +0.3 because

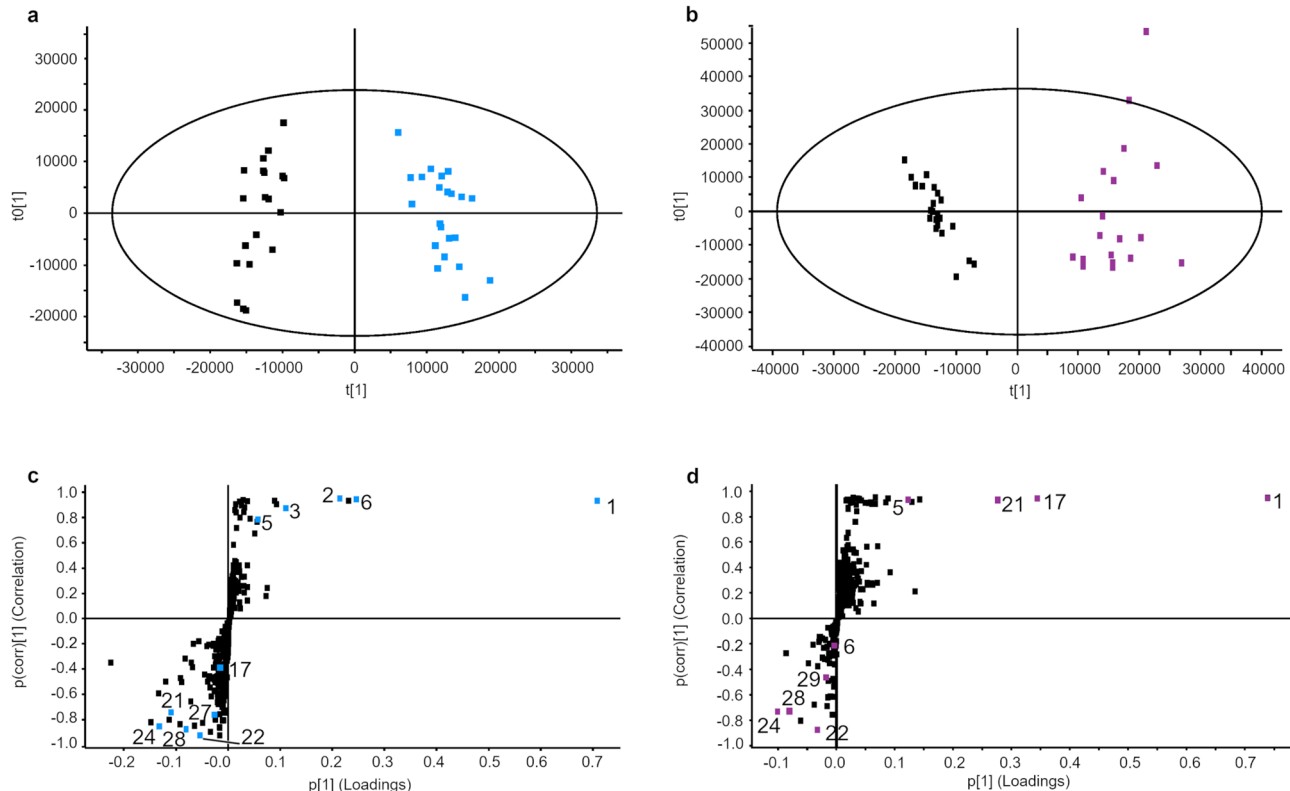

**Fig. 1 OPLS-DA scores plots and corresponding S-plots based on UHPLC/Q-TOF-MS profiling data of root extracts from VIGS and control lines.** The OPLS-DA scores plots are shown in **a**, **b** for *AbP450-5021* VIGS lines (blue) vs. control lines (black) (Pareto scaling, R2Y = 0.96, Q2Y = 0.94) and *AbP450-116623* VIGS lines (purple) vs. control lines (black) (Pareto scaling, R2Y = 0.94, Q2Y = 0.93), respectively. The data points in the OPLS-DA scores plots refer each to extracts of individual plants and are color-coded according to the genotype. The ellipse indicates 95% confidence. The covariance p[1] and correlation p(corr)[1] loadings from the two class OPLS-DA models are shown here in S-Plot format (**c** control = −1, *AbP450-5021* VIGS = 1 and **d** control = −1, *AbP450-116623* VIGS = 1). In the S-plots, each point represents a different exact mass-to-charge ratio/retention time pair. The upper right quadrant of the S-plots shows exact mass-to-charge ratio/retention time pairs whose levels are higher in VIGS lines than in control lines, whereas the lower left quadrant indicates mass-to-charge ratio/retention time pairs whose levels are higher in control lines. The further along the x-axis the greater the contribution to the variance between the groups, while the further the y-axis the higher the reliability of the analytical result. Data points for selected metabolites are highlighted: pseudotropine (**1**), acetyl pseudotropine (**2**), propionyl pseudotropine (**3**), 5:0-acyl pseudotropine (**5**), tigloyl pseudotropine (**6**), norpseudotropine (**17**), calystegine A5 (**27**), calystegine A3 (**28**), calystegine B1 (**29**), tigloyl norpseudotropine (**21**), tigloyl hydroxynorpseudotropine isomer 1 (**22**), and tigloyl dihydroxynorpseudotropine isomer 1 (**24**).

these boundaries retained signals with $P < 0.05$ in at least one genotype. To eliminate fragment ions and compounds without nitrogen, further filtering removed ions below mass-to-charge ratio ($m/z$) of $m/z$ 125 (mass of the tropane core is 125 Da), odd nominal $m/z$ which have zero or an even number of nitrogen atoms according to the nitrogen rule, and metabolites with $P$ values > 0.02 in both VIGS genotypes[21]. Forty-one alkaloids in *A. belladonna* root, most of them hydroxylated and/or esterified, remained after filtering and were annotated based on accurate mass measurements of positive ions observed in UHPLC/MS and UHPLC/MS/MS spectra (Supplementary Data 1), with all yielding $q$-values, a measure of false discovery rate, less than 0.05 providing they were resolved above the limit of quantification (S/N > 10)[22]. Levels of all 41 metabolites distinguished the *AbP450-5021* lines from respective controls with 23 distinguishing *AbP450-166623* from respective controls. Pseudotropine (**1**) and metabolites annotated as alkaloids with a pseudotropine core structure were assigned, including short-chain *O*-acyl pseudotropine esters with acyl chain lengths of two to five carbon atoms (**2**-**6**) and a di-acylated derivative (**7**). Neither tropine nor any of the known, major tropine-derived alkaloids from the scopolamine biosynthesis pathway exhibited levels that distinguish VIGS from control lines, but malonyl-phenyllactoyl (or malonyl-tropyl) tropanol (**8**) and various oxidized derivatives (**10**-**12**), three

hydroxytropanol isomers (**13**-**15**), and tropanol hexoside (**16**) which distinguished VIGS from control lines may include tropine derivatives yet to be characterized and reported (Supplementary Data 1). Additionally, the following metabolites with MS/MS spectra indicative of *N*-demethylated norpseudotropine scaffolds distinguished VIGS lines from their respective control lines: norpseudotropine (**17**), short-chain *O*-acyl norpseudotropine esters (**18**-**19**, **21**) and their hydroxylated derivatives (**22**-**24**, **26**), and calystegines (**27**-**29**) (Supplementary Data 1). Furthermore, nicotinic acid hexoside (**30**) and metabolites with $m/z$ 172.13 (**31**), $m/z$ 228.16 (**32**), $m/z$ 244.15 (**33**), $m/z$ 280.19 (**34**), $m/z$ 292.15 isomers 1 and 2 (**35**, **36**), $m/z$ 326.20 (**37**), $m/z$ 454.15 isomer 1 (**38**), and $m/z$ 454.20 isomer 2 (**41**) met the above described criteria in at least one of the VIGS genotype/control line data sets (Supplementary Data 1). The MS/MS spectra of $m/z$ 172.13 (**31**), $m/z$ 292.15 isomers 1 and 2 (**35**, **36**) and the hexosylated derivative $m/z$ 454.20 isomer 2 (**41**) are suggestive of alkaloids with five-membered ring heterocyclic cores. Exact masses and isotopolog abundances led to annotation of $m/z$ 454.15 isomer 1 (**38**) as a sulfur-containing phenyllactoyl tropanol derivative, but MS/MS spectra did not reveal information about the nature of the sulfur-containing moiety. Several metabolites were present in multiple isomeric forms with identical exact molecular and fragment ion masses and are numbered

according to their relative chromatographic elution order (Supplementary Data 1). Low abundance isomers of the above described alkaloids, 9, 25, 39 and 40, and metabolite 20 were not reported by automated data processing but were discovered by manual inspection of profiling data. Fold changes for these compounds were determined by manual peak integration, and P-values for each are significant in at least one VIGS genotype (Supplementary Data 1).

Notably, both VIGS genotypes were impaired in the production of various hydroxylated N-demethylated alkaloids with norpseudotropine cores (22-29) including calystegines whereas the levels of pseudotropine (1) were increased (Fig. 1c, d). Efficient silencing of a gene involved in a biosynthetic pathway will often, in the absence of isofunctional genes or an alternative biosynthetic route, cause a decrease in the levels of product(s) and metabolites further "downstream" from the affected reaction. This can be accompanied by increased accumulation of the substrate of the affected reaction and other "upstream" metabolites that are not efficiently channeled into the biosynthesis of "downstream" metabolites[16,23]. Thus, the observed decrease in hydroxylated N-demethylated alkaloids with norpseudotropine cores (22-29) and increase in N-methylated pseudotropine (1), a plausible "upstream" precursor, suggest both AbP450-5021 and AbP450-116623 catalyze reactions in the biosynthesis of N-demethylated modified tropane alkaloids. The analysis also revealed differences between the VIGS genotypes suggesting that suppression of the target transcript abundance affected distinct reactions in the VIGS genotypes (Fig. 1c, d).

### *AbP450-5021* and *AbP450-116623* VIGS lines are impaired in the biosynthesis of *N*-demethylated modified tropane alkaloids.

To further assess the impact of gene silencing in A. belladonna, we compared the relative levels of abundant metabolites and their isomers in roots of AbP450-5021 and AbP450-116623 VIGS lines to levels in respective control lines. We propose that mean fold-change in a metabolite level in VIGS genotypes of >25% (compared to control lines) together with statistical thresholds using P-value < 0.02 and q < 0.05 to be considered as significant. The fold-change threshold was established to be greater than three times the empirical mean coefficient of variation for levels of four metabolites expected to be minimally affected by silencing (tropine, scopolamine, littorine, and hyoscyamine; mean CV ~8%). Levels of tropine, littorine, hyoscyamine and scopolamine in both VIGS genotypes did not significantly differ from those in respective control lines based on statistical and effect size considerations (Supplementary Data 2). In contrast, the levels of the following nine annotated metabolites with N-methylated pseudotropine core structures were significantly increased at least 1.7-fold and up to 14-fold higher in roots of AbP450-5021 VIGS lines than in control lines: pseudotropine (1), short-chain O-acyl pseudotropine esters (2-6), hydroxytropanol isomers 2 and 3 (14, 15), and tropanol hexoside (16) (Fig. 2, Supplementary Data 1). The levels of 7:1-acyloxy tigloyloxy tropane (7), malonylphenyllactoyl tropanol isomers and oxidized derivatives (8-12), and hydroxytropanol isomer 1 (13) were decreased by 1.4-fold to 4-fold in these VIGS lines (Fig. 2, Supplementary Data 1). Furthermore, the levels of thirteen N-demethylated alkaloids with norpseudotropine core were significantly decreased in the AbP450-5021 VIGS genotype by at least 1.4-fold and up to 7-fold: norpseudotropine (17), short-chain O-acyl norpseudotropine esters (18-21) and mono-, di-, and tri-hydroxylated esters (22-26), and calystegines (27-29) (Fig. 2, Supplementary Data 1). In these VIGS lines, the levels of nicotinic acid hexoside (30) and metabolites m/z 172.13 (31), m/z 228.16 (32), m/z 244.15 (33), m/z 280.19 (34), m/z 292.15 isomers 1 and 2 (35, 36), m/z 326.20 (37), m/z 454.15 isomers 1 and 2 (38, 39), and m/z 454.20 isomers 1 and 2 (40, 41) decreased from 1.6-fold to

4-fold (Fig. 2, Supplementary Data 1). Hence, the decreased levels of metabolites with N-demethylated cores in AbP450-5021 VIGS lines and concomitant increased levels of metabolites with N-methylated cores including plausible "upstream" precursors, pseudotropine (1) and short-chain O-acyl pseudotropines (2-7), suggest that AbP450-5021 has N-demethylase activity critical for the production of the annotated alkaloids with norpseudotropine cores in A. belladonna (Fig. 2, Supplementary Data 1). For the phenyllactoyl and malonyl-phenyllactoyl esters 8, 9, 38, and 39, nicotinic acid hexoside (30), and the alkaloids 31, 35, 36, and hexosylated derivatives 40 and 41, a direct link to a cytochrome P450 activity is not evident according to structural information from MS/MS data (Supplementary Data 1). The decreased levels of these metabolites and those of derivatives (10-12) are likely indirect metabolic perturbations as a consequence of gene silencing in AbP450-5021 VIGS lines and attributed to mechanisms regulating the biosynthesis and/or conversion of these metabolites to other unknown chemical forms.

Similar to AbP450-5021 VIGS lines, roots of AbP450-116623 VIGS lines accumulated from 4-fold to 12-fold higher levels of pseudotropine (1), O-acyl pseudotropine esters with short-chain acyl groups (10-13) and hydroxytropanol isomers 2 and 3 (14, 15) relative to control lines. Levels of tigloyl pseudotropine (6) and hydroxytropanol isomer 1 (13) did not significantly differ from control lines (Fig. 2, Supplementary Data 1). With regard to N-demethylated alkaloids with a norpseudotropine scaffold, levels of norpseudotropine (17) and its short-chain acyl esters (18-21), which lack additional ring-hydroxyl groups, ranged from 8-fold to 100-fold higher in AbP450-116623 VIGS lines compared to control lines (Fig. 2, Supplementary Data 1). In contrast, levels of ring-hydroxylated tigloyl hydroxynorpseudotropine isomer 1 (22) and the further modified di- and trihydroxylated norpseudotropine core alkaloids tigloyl dihydroxynorpseudotropine isomers (24, 25), tigloyl trihydroxynorpseudotropine (26), and calystegines (27-29) were 1.8- to 3-fold decreased in AbP450-116623 VIGS lines relative to control lines (Fig. 2, Supplementary Data 1). The differences in the outcome of gene silencing on levels of norpseudotropine (17) and short-chain O-acyl norpseudotropine esters (18-21) between AbP450-116623 and AbP450-5021 VIGS lines (relative to control lines) point towards a role of AbP450-116623 "downstream" of AbP450-5021 in the biosynthesis of N-demethylated modified tropane alkaloids. Since the levels of further ring-hydroxylated norpseudotropine alkaloids, including the metabolites 22 and 24-26 representing hydroxylated products derived from tigloyl norpseudotropine (21), were decreased in AbP450-116623 VIGS lines, we postulated that AbP450-116623 is a ring-hydroxylase that catalyzes the conversion of tigloyl norpseudotropine (21) to tigloyl hydroxynorpseudotropine isomer 1 (22) (Fig. 2). The latter is likely further hydroxylated at the bicyclic ring to tigloyl dihydroxynorpseudotropine (24, 25) and tigloyl trihydroxynorpseudotropine (26) (Fig. 2). In control lines, tigloyl hydroxynorpseudotropine isomer 1 (22) was the major isomer, accumulating at 10-fold and higher levels than tigloyl hydroxynorpseudotropine isomer 2 (23). Our analyses revealed that in AbP450-116623 VIGS lines, the levels of tigloyl hydroxynorpseudotropine isomers 1 (22) and 2 (23) were unequally affected, and may be explained by differential conversions of isomers varying in hydroxyl group positions. In contrast to AbP450-5021 VIGS lines, levels in AbP450-116623 VIGS lines of nicotinic acid hexoside (30), m/z 280.19 (34), m/z 292.15 isomers (35, 36), m/z 454.15 isomer 1 (38), and m/z 454.20 isomer 2 (41) were not significantly affected compared to control lines. However, levels of m/z 172.13 (31), m/z 244.15 (33), and m/z 326.20 (37) were 2-fold decreased and correlated with decreased AbP450-116623 gene expression in the AbP450-116623 VIGS lines (Fig. 2, Supplementary Data 1).

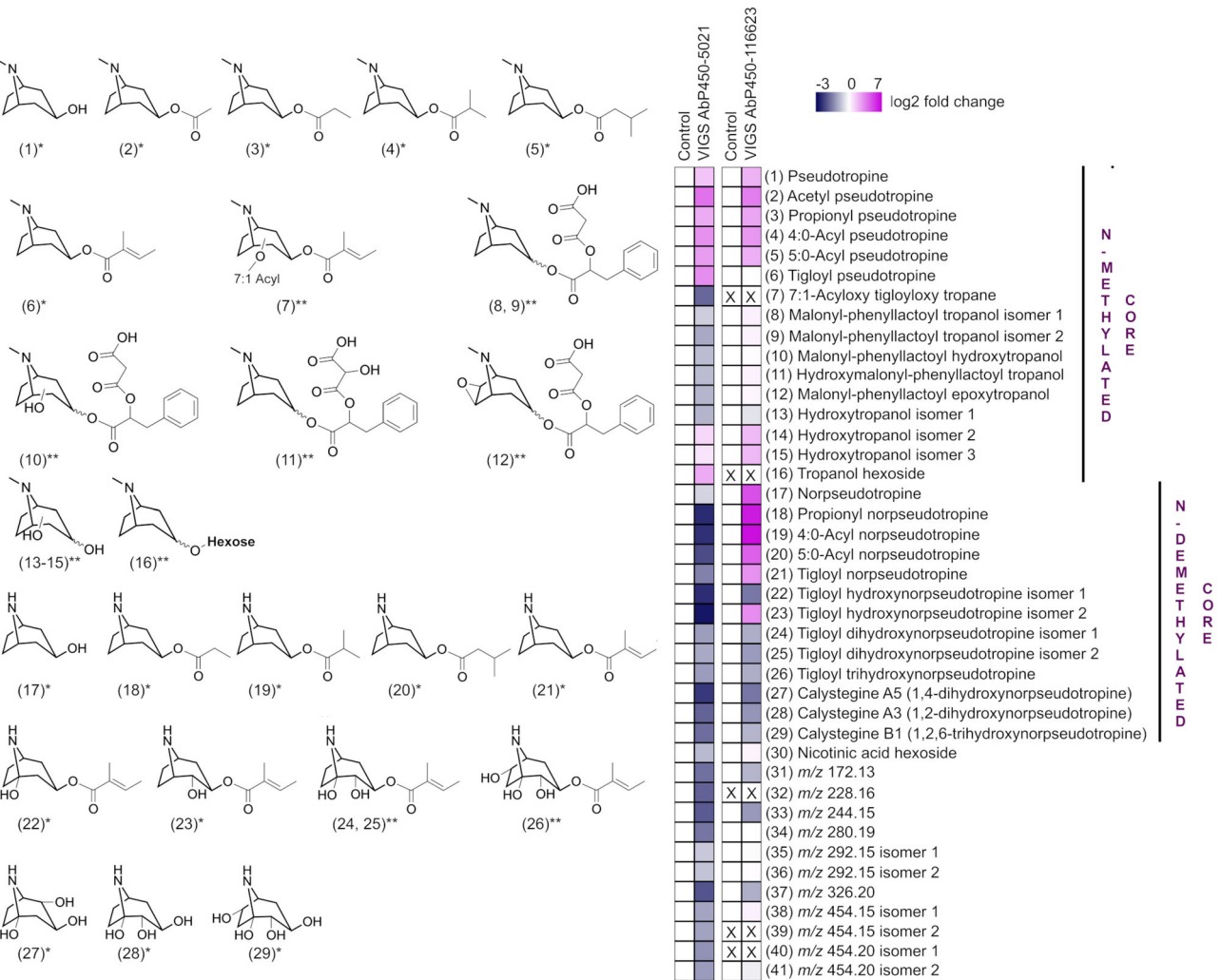

**Fig. 2 Metabolic changes in roots of *AbP450-5021 VIGS* and *AbP450-116623 VIGS* lines relative to control lines.** Structures of annotated alkaloids with pseudotropine and norpseudotropine cores are shown (*, authentic reference; **, putative structure based on UHPLC/MS/MS data, reported in Supplementary Data 1). Heat maps depict log2-fold changes in the metabolite levels in VIGS lines relative to control lines (white, control level; gray-blue, decreased level; purple: increased level; X: below limit of detection S/N < 3). Source data are provided as a Source Data file.

**AbP450-5021 and AbP450-116623 catalyze sequential reactions in the biosynthesis of modified alkaloids with a norpseudotropine scaffold.** To investigate the catalytic activities of AbP450-5021 and AbP450-116623, *in planta* biochemical assays were conducted using an *Agrobacterium*-mediated transient expression system in leaves of *N. benthamiana*. In our UHPLC/HRMS analyses, extracts of wild-type *N. benthamiana* leaves lacked metabolites with tropine, pseudotropine, and *N*-demethylated core structures that were detected in *A. belladonna* (Supplementary Data 1). Consistent with these findings, phylogenetic analyses did not identify any obvious candidate orthologs for AbP450-5021 and AbP450-116623 in *N. benthamiana* genome assemblies. However, further analyses did identify genes encoding highly conserved AbP450-5021 and AbP450-116623 candidate orthologs in various other members of the Solanaceae that produce calystegines, including in the major vegetable crops tomato, potato, eggplant, and pepper (Supplementary Figs. 4 and 5)[2,24].

In our assay setup, leaves of *N. benthamiana* were first infiltrated to produce AbP450-5021 and AbP450-116623 alone or AbP450-5021 + AbP450-116623 together and then infiltrated with candidate substrates to assay for *N*-demethylase and ring-hydroxylase activity. Short-chain *O*-acyl pseudotropine and

*O*-acyl tropine esters with acetyl (2:0-acyl), propionyl (3:0-acyl), isobutyryl (4:0-acyl), isovaleryl (5:0-acyl), senecioyl (5:1-acyl) and tigloyl (5:1-acyl) groups, and tigloyl norpseudotropine were chemically synthesized as candidate substrates, and their structures were verified by mass spectrometry and NMR spectroscopy (Supplementary Data 3, Supplementary Data 6). In addition, authentic non-acylated tropinone, tropine, nortropine, pseudotropine, and norpseudotropine were tested as candidate substrates. Control experiments were conducted with infiltrated *N. benthamiana* leaves that did not produce any *A. belladonna* cytochrome P450s (empty vector control), and metabolite extracts from infiltrated *N. benthamiana* leaves were analyzed by untargeted UHPLC/HRMS. Only leaves producing either AbP450-5021 alone or AbP450-5021 + AbP450-116623 yielded evidence of conversion of short-chain *O*-acyl pseudotropine esters to respective *N*-demethylated *O*-acyl norpseudotropine products (Fig. 3, Supplementary Figs. 6–10, Supplementary Data 3 and 4). No products of *N*-demethylase activity were detected in AbP450-5021 or AbP450-5021 + AbP450-116623-producing leaves that were assayed with acetyl pseudotropine, short-chain *O*-acyl tropine esters or any of the non-acylated *N*-methylated candidate substrates (tropinone, tropine, and pseudotropine). Control leaves and leaves with AbP450-116623

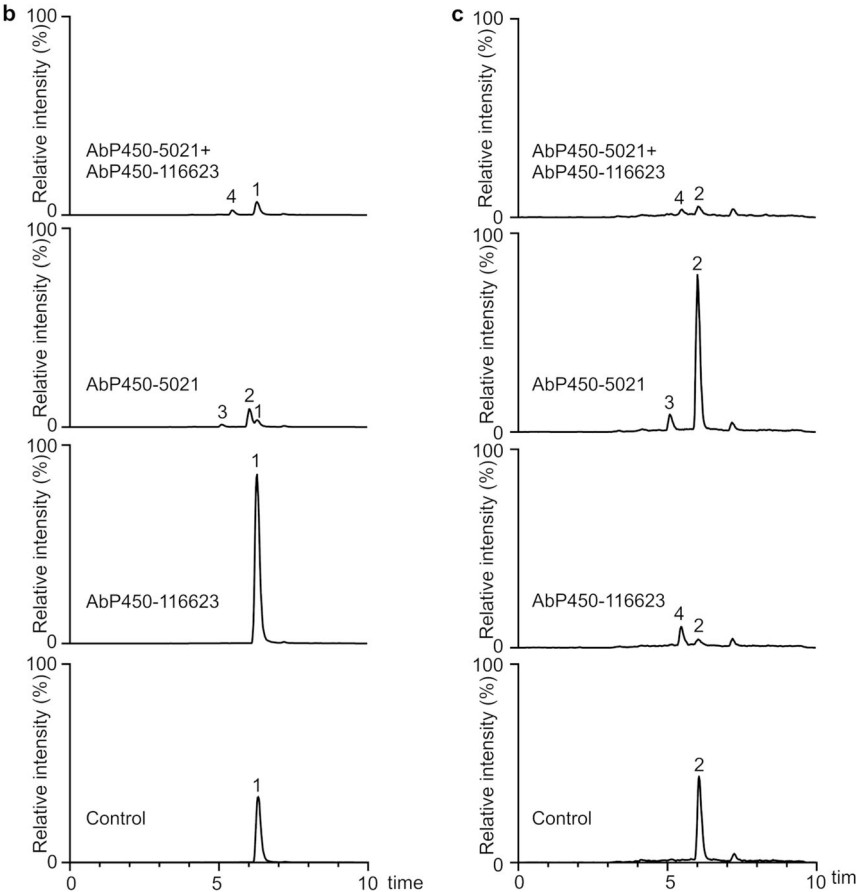

**Fig. 3 N. benthamiana transient assays with short-chain O-acyl pseudotropines.** N. benthamiana leaves producing AbP450-5021, AbP450-116623 or AbP450-5021 + AbP450-116623 were assayed with short-chain O-acyl pseudotropine esters for N-demethylase and ring-hydroxylase activity (**a**) (-, no products detected; blue circle, N-demethylase product O-acyl norpseudotropine; purple circle, ring-hydroxylase product O-acyl hydroxynorpseudotropine). Control experiments were conducted with infiltrated leaves that did not produce A. belladonna cytochrome P450s. Leaf metabolite extracts were analyzed by UHPLC/MS using a 10-min gradient elution method (Supplementary Data 3 and 4). Each condition was tested in triplicate, and the experiments were replicated twice. Representative chromatograms are shown for assays with N-methylated tigloyl pseudotropine (**b**) and N-demethylated tigloyl norpseudotropine (**c**). Each chromatogram represents a combined extracted ion chromatogram for the exact masses of tigloyl pseudotropine (1), tigloyl norpseudotropine (2), tigloyl hydroxynorpseudotropine isomer 2 (3) and tigloyl hydroxynorpseudotropine isomer 1 (4). Note the absolute vertical scale (y-axis) is identical for each set of chromatograms shown in **b**, **c**.

alone did not yield demethylated products with any of the N-methylated candidate substrates tested (Fig. 3, Supplementary Figs. 6–10, Supplementary Data 3 and 4). In leaves producing AbP450-5021 + AbP450-116623, short-chain O-acyl pseudotropine substrates with acyl chain lengths of three to five carbon atoms were consumed and, depending on the substrate used, products matching the respective N-demethylated short-chain O-acyl hydroxynorpseudotropine esters accumulated (Fig. 3, Supplementary Figs. 6–10, Supplementary Data 3 and 4). Hydroxylated products were neither detected in assays with AbP450-5021 + AbP450-116623 following infiltration of short-chain O-acyl tropine esters nor when AbP450-5021 or AbP450-116623 were each separately tested in planta with short-chain O-acyl tropine esters or saturated O-acyl pseudotropine esters. Assays with AbP450-116623 alone and N-demethylated tigloyl norpseudotropine as substrate yielded tigloyl hydroxynorpseudotropine (isomer 1), providing evidence that AbP450-116623 is a ring-hydroxylase that uses N-demethylated products from

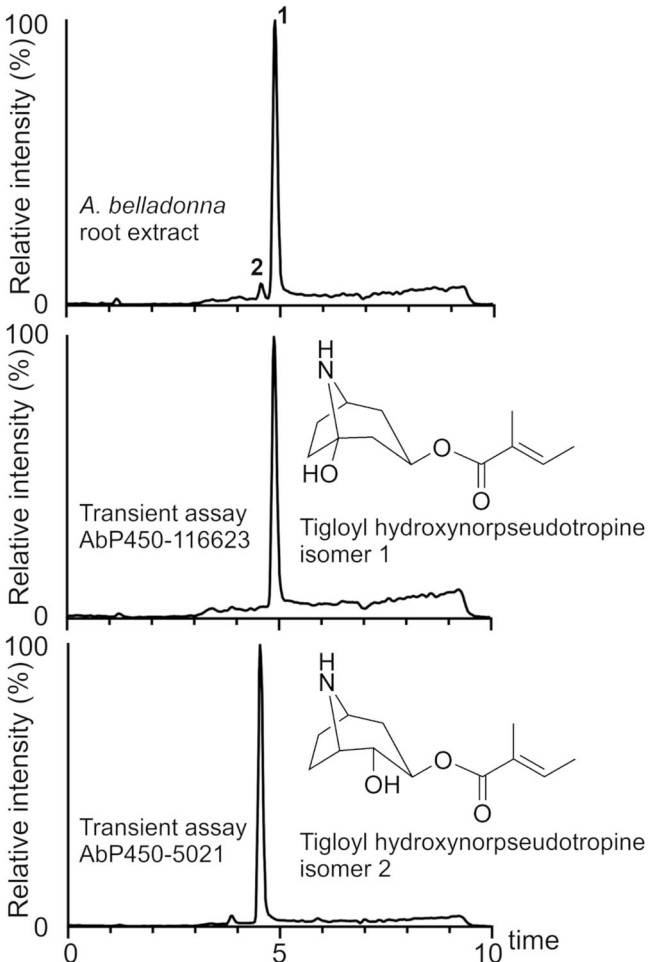

**Fig. 4 Comparative analysis of tigloyl hydroxynorpseudotropine isomers produced in *A. belladonna* root and *N. benthamiana* transient assays.** *N. benthamiana* leaves producing AbP450-5021 and AbP450-116623 were assayed with tigloyl norpseudotropine for ring-hydroxylase activity. Metabolite extracts were analyzed by UHPLC/MS using a 10-min gradient elution method. Each chromatogram represents an extracted ion chromatogram for *m/z* 226.14. NMR confirmed structures are shown for tigloyl hydroxynorpseudotropine isomers 1 and 2 that were purified from up-scaled transient assays (this study, Supplementary Data 6). 1, *A. belladonna* tigloyl hydroxynorpseudotropine isomer 1 (**25**); 2, *A. belladonna* tigloyl hydroxynorpseudotropine isomer 2 (**26**).

AbP450-5021-catalyzed reactions as substrates (Fig. 3, Supplementary Data 3 and 4). Note that in assays with AbP450-5021 and AbP450-116623 each alone or AbP450-5021 + AbP450-116623 together, conversion of *N*-methylated *O*-acyl pseudotropines, *O*-acyl tropines, pseudotropine, tropine, and tropinone to ring-hydroxylated *N*-methylated products with one and up to three additional hydroxyl groups was not detected.

**Both AbP450-5021 and AbP450-116623 catalyze ring-hydroxylations of tigloyl norpseudotropine to tigloyl hydroxynorpseudotropine isomers.** Transient assays with *N. benthamiana* leaves producing AbP450-5021 alone using either tigloyl pseudotropine or tigloyl norpseudotropine as substrate also yielded a hydroxylated product whose mass spectrum matched that of the AbP450-116623 product tigloyl hydroxynorpseudotropine (isomer 1) (Fig. 3, Supplementary Data 3 and 4). However, the hydroxylated AbP450-5021 product eluted at a different retention time in UHPLC/HRMS analyses

compared to the AbP450-116623 product (Fig. 3b, c). We refer to these compounds hereafter as tigloyl hydroxynorpseudotropine isomer 1 for the AbP450-116623 product and tigloyl hydroxynorpseudotropine isomer 2 for the AbP450-5021 product. To assess the nature of the two isomers, they were purified from up-scaled *N. benthamiana* transient assays and characterized using a series of NMR spectroscopic experiments. The analyses confirmed they are tigloyl hydroxynorpseudotropine isomers differing in the position of hydroxylation on the bicyclic ring (Fig. 4, Supplementary Data 6). In this context, it should be noted that for tigloyl hydroxynorpseudotropine isomer 1, no $^{13}$C NMR signal was detected for the carbon atom next to the ring nitrogen as the molecules undergo rapid tautomerism between the bicyclo-hemiaminal and the seven-membered ring amino ketone during NMR measurement as known to occur with 1-hydroxy(nor)tro-pane derivatives (Supplementary Data 6)[25]. However, all other signals were unequivocally assigned to the corresponding carbon positions in the structure and consistent with tigloyl hydroxynorpseudotropine isomer 1 (AbP450-116623 product) being hydroxylated at a carbon atom adjacent to the ring nitrogen (Fig. 4, Supplementary Data 6). In contrast, tigloyl hydroxynorpseudotropine isomer 2 (AbP450-5021 product) has an equatorial hydroxyl group at a ring carbon adjacent to the esterified carbon atom based on $^3J_{HH}$ coupling constant analysis of NMR spectra (Fig. 4, Supplementary Data 6). Each of these isomers may exist as two mirror-image enantiomers that NMR spectroscopic analyses cannot distinguish because their NMR spectra would be identical.

UHPLC/HRMS analyses of *A. belladonna* root metabolite extracts together with the chemically synthesized alkaloids and products in extracts from *N. benthamiana* transient assays provided further evidence for the annotation of short-chain *O*-acyl pseudotropine (**2**-**6**) and *O*-acyl norpseudotropine (**18**-**21**) esters detected in *A. belladonna* root (Fig. 2, Supplementary Data 1). In comparative UHPLC/HRMS and UPHLC/MS/MS analyses, retention times and MS/MS spectra of tigloyl hydroxynorpseudotropine isomer 1 (AbP450-116623 product) and isomer 2 (AbP450-5021 product) from transient assays in *N. benthamiana* matched tigloyl hydroxynorpseudotropine isomers 1 (**22**) and 2 (**23**) in *A. belladonna* root extracts, respectively (Fig. 4, Supplementary Data 1 and 3). These data are consistent with the metabolic phenotype of *A. belladonna* AbP450-116623 VIGS lines in which the decreased level of tigloyl hydroxynorpseudotropine isomer 1 (**22**) correlates with AbP450-116623 gene silencing, while the production of tigloyl hydroxynorpseudotropine isomer 2 (**23**) was increased relative to control lines and inversely correlated with AbP450-116623 gene silencing (Fig. 2). Taken together, our data provide evidence that tigloyl hydroxynorpseudotropine isomers 1 (**22**) and 2 (**23**) in *A. belladonna* are products from AbP450-116623 and AbP450-5021-catalyzed reactions, respectively.

## Discussion

The discovery metabolomics-guided approach taken in the present study revealed involvement of two cytochrome P450 genes, *AbP450-5021* and *AbP450-116623*, in an intricate and dynamic alkaloid sub-network in *A. belladonna* root that channels pseudotropine into the production of modified tropane alkaloids. The findings led us to develop a model for the pseudotropine-dependent alkaloid sub-network in *A. belladonna* root (Fig. 5). AbP450-5021 has a central role as *N*-demethylase/ring-hydroxylase in the early reaction steps during the biosynthesis of all discovered *N*-demethylated modified tropane alkaloids with norpseudotropine cores. Oxidative *N*-demethylation catalyzed by cytochrome P450s proceeds via the insertion of a single oxygen

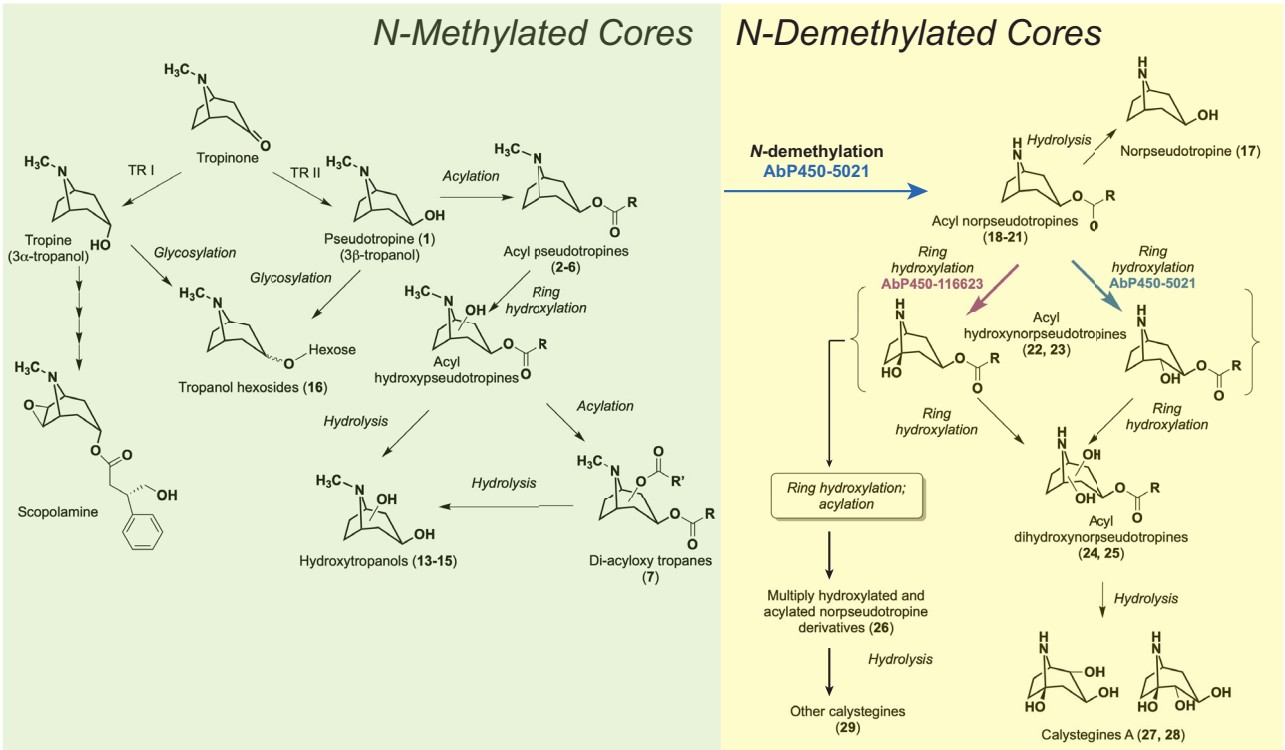

**Fig. 5 Model for the pseudotropine-dependent biosynthesis of modified tropane alkaloids in *A. belladonna* root and the roles of AbP450-5021 and AbP450-116623.** The depicted alkaloid sub-network model is based on our findings from discovery metabolite profiling in *A. belladonna* VIGS and control lines, transient assays in *N. benthamiana*, and NMR experiments. Note annotated metabolites supporting this model are indicated by numbers (Supplementary Data 1, Fig. 2). Pseudotropine is converted to acyl pseudotropines that are channeled into the production of acylated and non-acylated modified tropane alkaloids with *N*-methylated and *N*-demethylated cores. The biosynthesis of *N*-demethylated alkaloids with norpseudotropine cores depends on the activity of the *N*-demethylase/ring-hydroxylase AbP450-5021. This enzyme catalyzes the *N*-demethylation of acyl pseudotropines to acyl norpseudotropines. Norpseudotropine is likely formed through hydrolysis of acyl norpseudotropines. Both AbP450-116623 (ring-hydroxylase) and AbP450-5021 (*N*-demethylase/ring-hydroxylase) catalyze the subsequent conversion of acyl norpseudotropines to respective acyl hydroxynorpseudotropines. Note that acyl norpseudotropines are *meso* compounds each with a mirror plane of symmetry. Thus, positions 1 and 5 (carbon atoms adjacent to the ring-nitrogen) are equivalent as are the positions 2 and 4 (carbon atoms adjacent to the esterified carbon atom), yielding the potential to form, for example, acyl hydroxynorpseudotropine enantiomers that are hydroxylated in either position 1 or 5. Acyl hydroxynorpseudotropines are subject to further modifications, and the resulting metabolites are likely metabolized to non-acylated calystegines.

atom into a C-H bond of the methyl group and subsequent spontaneous conversion of the hydroxylated intermediate into the *N*-demethylated product and the by-product formaldehyde[26]. Thus, AbP450-5021's role in *N*-demethylation derives from its hydroxylase activity with acylated substrates that have the 3β-configuration of the exocyclic oxygen at the *N*-methylated bicyclic ring structure (pseudotropine core). It is noteworthy that AbP450-5021 lacked activity with acetyl pseudotropine in transient assays, consistent with the root metabolite profile in *A. belladonna* where acetyl norpseudotropine was not detected, but norpseudotropine esters of three to five carbons (**18-21**) were observed (Fig. 3, Supplementary Data 1). This suggests that an acyl chain with at least three carbon atoms may be necessary to orient the substrate in the active site of AbP450-5021 for hydroxylation initiating *N*-demethylation. Ring-hydroxylation is also catalyzed by AbP450-5021 at the carbon atom adjacent to the esterified carbon atom at the 3-position of tigloyl norpseudotropine (isomer 2), whereas AbP450-116623 catalyzes ring-hydroxylation at the carbon atom adjacent to the ring nitrogen of *O*-acyl norpseudotropines with various acyl chain lengths (Figs. 4 and 5, Supplementary Data 6). In *A. belladonna* root extracts, levels of *O*-acyl hydroxynorpseudotropine products with non-tigloyl short chain acyl groups were below detection limit. They may be metabolized, or native AbP450-116623 may have no or very limited access to *O*-acyl norpseudotropine substrates with

an acyl group other than tigloyl (5:1-acyl) in *A. belladonna* root. Future investigations will show whether AbP450-116623 and/or AbP450-5021 also contribute through ring-hydroxylation to the production of the as yet unidentified metabolites **32**-**34** and **37**. The alkaloids **31**, **35**, **36**, **40**, and **41** lack a bicyclic heterocyclic ring-system, and the MS/MS data do not suggest that their heterocyclic cores are hydroxylated or otherwise oxidized.

Products of AbP450-5021-catalyzed reactions are subject to further modifications (Fig. 5). The findings that in transient assays, AbP450-5021, AbP450-116623, and AbP450-5021 + AbP450-116623 reproducibly failed to catalyze modifications of non-acylated candidate substrates, including tropine, pseudotropine, and norpseudotropine, have several implications. With regard to the *N*-methylated hydroxytropanol isomers 1-3 (**13-15**), assay data together with the differing trends in abundance in VIGS lines relative to control lines, indicate that distinct competing hydroxylases and/or hydrolases are involved in their biosynthesis in *A. belladonna*. For example, hydroxytropanols may be produced through hydrolysis of *N*-methylated acyl hydroxypseudotropines (Fig. 5). The sub-network model also takes into consideration that *N*-methylated tropanol hexosides may be derived from both pseudotropine and tropine (Fig. 5). Since AbP450-5021 does not mediate the *N*-demethylation of pseudotropine, but the norpseudotropine level was decreased in AbP450-5021 VIGS lines, it is plausible that norpseudotropine (**17**) is produced via hydrolysis of

*N*-demethylated *O*-acyl norpseudotropine intermediates (**18-21**) in *A. belladonna* (Figs. 2 and 5). Several observations taken together indicate that direct conversion of norpseudotropine (**17**) into calystegines (**27-29**) via a sequence of hydroxylation steps is unlikely. In our analyses, metabolite extracts from *A. belladonna* root lacked detectable levels of hydroxynorpseudotropine isomers that would represent plausible intermediates for the production of the dihydroxylated norpseudotropine alkaloids calystegines A5 (**27**) and A3 (**28**). Calystegine biosynthesis appeared to depend on both AbP450-5021 and AbP450-116623 activities as the respective VIGS lines were impaired in the production of calystegines A5 (**27**), A3 (**28**) and B1 (**29**) (Fig. 2). However, AbP450-5021, AbP450-116623, and AbP450-5021 + AbP450-116623 showed no activity in transient assays with non-acylated candidate substrates including pseudotropine or norpseudotropine, but both AbP450-5021 and AbP450-116623 exhibited activity with acylated substrates (Fig. 3, Supplementary Figs. 6–10, Supplementary Data 3 and 4). It is, therefore, likely that AbP450-5021 and AbP450-116623 contribute to the production of *O*-acyl hydroxynorpseudotropine intermediates that are further hydroxylated prior to being metabolized to calystegines (Fig. 5). This is consistent with structural features of the calystegines A5 (**27**), A3 (**28**) and B1 (**29**) that for example, share the *N*-demethylated norpseudotropine core structure and a hydroxyl group at position 1 next to the nitrogen atom (Figs. 2 and 5). In fact, *A. belladonna* accumulates alkaloids matching tigloyl dihydroxynorpseudotropine (**24**, **25**) and tigloyl trihydroxynorpseudotropine (**26**) that upon hydrolysis would yield calystegines A (dihydroxynorpseudotropine isomers) and B (trihydroxynorpseudotropine), respectively (Figs. 2 and 5). These findings suggest that enzymes that catalyze acylation and acyl group removal via hydrolysis may have important roles in calystegine biosynthesis (Fig. 5).

Studies on the genus *Datura* (Solanaceae family), in particular the species *Datura stramonium* and *Datura innoxia*, described the occurrence of hydroxylated and acylated alkaloids with *N*-methylated tropane cores[27–31]. Tigloyl pseudotropine (tigloidine) detected in *A. belladonna* (**6**) was previously identified in the Solanaceae genera *Datura*, *Physalis*, *Duboisia*, *Anthocercis*, and *Hyoscyamus*[27–30,32,33]. It is an analog of atropine with anticholinergic activity influencing the central nervous system[34]. Therefore, it is conceivable that other alkaloids from *A. belladonna*, especially hydrophobic tertiary amines with one or more acyl groups, can cross the blood-brain barrier and cause central nervous system effects in herbivores. With the exception of tigloyloxy nortropane, the literature on *Datura* does not report any *N*-demethylated short-chain acyloxy nortropane alkaloids in roots, probably due to their lower abundance compared to tropane alkaloids and limited detection sensitivity of the gas-chromatography/mass spectrometry-based approaches used[35–37]. The non-acylated calystegines A5, A3 and B1 are, together with calystegines B2 and B3, the most widely reported calystegines in plants. Since their initial discovery in root exudates of *Calystegia sepium* (Convolvulaceae) 30 years ago, calystegines were found to occur in diverse plant species of the Convolvulaceae, Erythroxylaceae, Brassicaceae, Moraceae, and Solanaceae plant families, including in *A. belladonna* and the major crop plants tomato, potato, eggplant, pepper, and sweet potato[2,3,38–42]. As polyhydroxylated alkaloids with an *N*-demethylated norpseudotropine scaffold, calystegines are too hydrophilic to pass the blood-brain barrier. Their structures rather resemble the pyranose form of monosaccharides and indeed, calystegines are sugar-mimic defense compounds that bind and inhibit glycosidases from mammals, plants and insects[24,43–45]. The presence of genes encoding highly-conserved homologs for AbP450-5021 (CYP71 family) and AbP450-116623 (CYP82 family) in members of the genera *Atropa*, *Datura*, *Solanum*, *Capsicum*, and *Petunia* is

consistent with the reported occurrence of *N*-demethylated modified tropane alkaloids and suggests that early steps in their biosynthesis are likely conserved across the Solanaceae. Literature searches did not reveal any other functionally characterized member of the CYP71 family that catalyzes, like AbP450-5021, a demethylation reaction (Supplementary Fig. 4). Most characterized members of the CYP82 family act, like AbP450-116623, as hydroxylases (Supplementary Fig. 5). With the exception of tropinone synthase (CYP82M3) from *Atropa belladonna* and *N*-demethylases (CYP82E4v1, CYP82E4v2, CYP82E21, CYP82E5v2, CYP82E10) from *Nicotiana tabacum* involved in pyridine/pyrrolidine alkaloid metabolism, no other cytochrome P450 from the Solanaceae family with roles in alkaloid biosynthesis were previously reported in the CYP71 and CYP82 families[16,46,47].

The metabolomics approach used in this study opens the door to understanding the diversity and biosynthesis of medicinal alkaloids in plants on network level including the plants' responses to metabolic shifts. In this study, metabolic perturbations in the VIGS lines led also to discovery of various "off-target" alkaloids. In particular, silencing of the gene encoding AbP450-5021 (*N*-demethylase/ring-hydroxylase) appears to be associated with more generic changes in alkaloid metabolism. These findings raise questions about regulatory mechanisms governing alkaloid production with potential importance for future engineering approaches.

## Methods

**VIGS in *A. belladonna***. Seeds of *A. belladonna* were surface-sterilized for 5 min with 50% commercial bleach solution containing 0.04% Tween 20 and rinsed six times with sterile distilled water. One volume of seeds was incubated in 10 volumes of gibberellic acid solution (0.1% gibberellic acid with ≥90% being gibberellin $A_3$ and 5% ethanol in water) overnight in the dark at room temperature. The gibberellic acid solution was discarded, and the seeds were transferred onto sterile, water-soaked filter paper in petri dishes for germination. Plates were incubated for approximately 10 days under a 16-h photoperiod at 100 µmol m$^{-2}$ s$^{-1}$ light intensity at room temperature until radicles emerged. The seeds were planted into Sunshine® Redi-Earth Plug and Seedling Mix, and seedlings were grown for approximately 10 days in a growth room under a 16-h photoperiod with 120 µmol m$^{-2}$ s$^{-1}$ light intensity at 22 °C. The two-component tobacco rattle virus (TRV)-based vector system was used for VIGS in *A. belladonna*[48,49]. Fragments of *AbP450-5021* (MH292965) and *AbP450-116623* (MH292967) were each amplified from cDNA synthesized from *A. belladonna* lateral-root RNA using gene-specific primer pairs and inserted into the *Pst*I-linearized pTRV2-LIC vector by ligation independent cloning (Supplementary Data 5). The identity of the constructs was confirmed by PCR and DNA sequencing. The constructs were introduced into *Agrobacterium tumefaciens* GV3101. Bacterial clones harboring pTRV1, pTRV2 empty vector (control) or a pTRV2-VIGS construct were grown in selective Luria-Bertani liquid medium for 16 h at 30 °C. The starter cultures were diluted 1:25 in induction medium (9.76 g L$^{-1}$ MES, 5 g L$^{-1}$ glucose, 0.24 g L$^{-1}$ NaH$_2$PO$_4$, 2 g L$^{-1}$ NH$_4$Cl, 0.6 g L$^{-1}$ MgSO$_4$·7H$_2$O, 0.3 g L$^{-1}$ KCl, 0.02 g L$^{-1}$ CaCl$_2$, 0.005 g L$^{-1}$ FeSO$_4$·7H$_2$O) and shaken for additional 24 h at 30 °C. Cells were then sedimented at 2500 x *g* for 10 min, washed in one volume infiltration buffer (10 mM MgCl$_2$ in 10 mM MES-KOH pH 5.6) and resuspended in one half-volume of infiltration buffer. The optical density at 600 nm (OD$_{600}$) of the suspensions was adjusted to 0.3. Acetosyringone was added to the bacterial suspension with pTRV1 in a final concentration of 400 µM. Equal volumes of the *A. tumefaciens* pTRV1 suspension and a pTRV2-construct harboring suspension were mixed. Cotyledons of approximately 3-week old *A. belladonna* seedlings, prior to emergence of the first true leaf, were infiltrated with suspension using a needleless syringe. For comparison, control experiments were conducted in parallel with seedlings from the same batch but infiltrated with a suspension of pTRV1 and pTRV2 empty vector. Substantial biological replication was built into the experimental design to add statistical power: *N* = 23 for AbP450-5021 VIGS lines and *N* = 23 for respective control lines; *N* = 23 for AbP450-116623 VIGS lines and *N* = 24 for respective control lines. The seedlings were re-transferred to the growth chamber and grown under the same conditions. Four weeks post-infiltration, whole roots were harvested from the *A. belladonna* VIGS and control lines, frozen in liquid nitrogen and stored at −80 °C.

**Real-time quantitative RT-PCR**. Total RNA was extracted from root tissue of *A. belladonna* plants using the E.Z.N.A. Plant RNA Kit combined with on-column DNase treatment (Omega Bio-Tek). The cDNAs were synthesized from 1 µg of total RNA using the SuperScript III first-strand synthesis kit (Invitrogen, Thermo Fisher Scientific). qRT-PCRs were performed in triplicate (technical replicates) in a

CFX384 Touch™ Real-Time PCR Detection System with a C1000 Touch™ Thermal Cycler (BioRad). Each 10-μL reaction contained 5 μL FAST SYBR master mix (Applied Biosystems), 300 nM forward and reverse primer (Supplementary Data 5), and 20 ng and 50 ng cDNA for *AbP450-5021* and *AbP450-116623* expression analysis, respectively. The following temperature profile was used: 2 min at 50 °C and 10 min at 95 °C, followed by 40 cycles of 15 s at 95 °C and 1 min at 60 °C. Data were analyzed using the CFX Manager™ software (Bio-Rad) to calculate average threshold cycle (CT) values and Δ$^{CT}$ mean for the target genes. The efficiency of gene silencing was assessed by real-time quantitative RT-PCR on root tissue from six *A. belladonna* VIGS lines in comparison to six control lines (biological replicates, *N* = 6). These lines were selected based on their median calystegine A5 and A3 abundance within the set of analyzed lines for each genotype. The data were normalized to *AbEF-1* (JX154677) transcripts, and $2^{-\Delta\Delta CT}$ were calculated to determine relative transcript levels of the target gene.

**Discovery metabolite profiling in *A. belladonna* root.** Frozen root tissue samples from *A. belladonna* VIGS and control lines were ground to fine powder. For each sample, 50 mg root tissue and 500 μL extraction solvent containing 20% methanol, 0.1% formic acid and 4 μM telmisartan (internal standard) in water were added, vortexed for 5 s and incubated on an orbital shaker for 3 h at room temperature in the dark. The samples were centrifuged at maximum speed for 15 min, and supernatants were transferred into fresh vials. For discovery metabolomics analyses, the root extracts were diluted 1:20 in acetonitrile to facilitate chromatographic separations. A 10-μL volume of each extract was analyzed using a Waters Xevo G2-XS Q-TOF mass spectrometer equipped with a Shimadzu LC-20AD HPLC system and SIL-5000 autosampler. Metabolites were separated on a COR-TECS hydrophilic interaction liquid chromatography (HILIC) column (Waters, silica stationary phase, 2.1 × 100 mm, 1.6 μm particle size) at 40 °C column temperature with mobile phases consisting of 100 mM ammonium formate and 1% formic acid in water (solvent A) and acetonitrile (solvent B) using either a 30-min gradient or a 14-min gradient elution method. The 30-min gradient employed an initial condition of 95% B from 0.00 to 2.00 min, linear gradient at 85% B at 3.00 min, linear gradient to 82% B at 15.00 min, linear gradient to 78% B at 25.00 min, linear gradient to 40% B at 26.00 min and a hold from 26.01 to 27.00 followed by a return to 95% B and hold from 27.01 to 30.00 min at a flow rate of 0.3 mL/min. The 14-min gradient was as follows: 95% B from 0.00 to 2.00 min, linear gradient to 85% B at 3.00 min, linear gradient to 78% B at 10.00 min, linear gradient to 40% B at 11.00 min and hold to 12.00 min followed by a return to 95% B and hold from 12.01 to 14.00 min at a flow rate of 0.3 mL/min. The mass spectrometer was equipped with an electrospray ionization source and operated at 3000 V capillary voltage, 40 V cone voltage, 100 °C source temperature, 350 °C desolvation temperature, 20 L/h cone gas flow, and 500 L/h desolvation gas flow. For metabolite profiling, mass spectrum acquisition was performed in positive ion mode over *m/z* 50 to 1000 with scan time of 0.25 s using centroid data acquisition and quasisimultaneous acquisition of low- and high-energy spectra (MS$^E$) using a collision energy ramp from 20 to 80 V for the latter. Leucine enkephalin [M + H]$^+$ was used as lockmass. Accurate masses of pseudomolecular and fragment ions were confirmed in UHPLC/MS/MS analyses using the following 90-min gradient: 95% B at 0.00 to 5.00 min, linear gradient to 85% B at 30.00 min, linear gradient to 80% B at 80.00 min, linear gradient to 40% B at 85.00 min, return to 95% B and hold from 85.01 to 90.00 min at a flow rate of 0.3 mL/min. To determine the relative levels of littorine and hyoscyamine, isomers not fully resolved using the gradients described above, extracts were analyzed using the following 15-min gradient: 95% B at 0.00 to 2.00 min, linear gradient to 85% B at 3 min, linear gradient to 78% B at 12.00 min, linear gradient to 40% B at 13.00 min, hold to 14.00 min, return to 95% B and hold from 14.01 to 15.00 min at a flow rate of 0.3 mL/min. MS/MS spectra were acquired for selected precursor *m/z* values in centroid mode with a collision energy ramp from 10 to 50 V, and leucine enkephalin lockmass correction (*m/z* 556.2766). Metabolites were annotated based on accurate mass measurements of positive ions and fragments observed in UHPLC/MS/MS mass spectra. Extracted ion chromatograms for each target analyte were integrated using QuanLynx tool (Waters) with a mass window allowance of 0.03. A signal-to-noise ratio of 10 was considered as the low limit for quantification. All calculated peak areas were normalized to the peak area for the internal standard telmisartan and tissue dry weight (Source Data).

**Multivariate analysis of discovery metabolomic data.** UHPLC/MS data obtained with the 30-min gradient elution method on *A. belladonna* VIGS and control lines were processed using Progenesis QI v2.4 software (Nonlinear Dynamics Ltd., Newcastle, UK). All retention times were aligned to a reference run selected by the software within each data set. Runs with an alignment score >85% were included for further analyses. The peak picking parameters were set to 0.02% base peak sensitivity level, 0.02 min minimum chromatographic peak width, and 1 to 25 min retention time limit. M + H, M + 2H, M + H-H₂O, M + NH₄, M + Na, M + K, 2 M + H, 2 M + NH₄, 2 M + Na and 2 M + K were considered as possible adduct ions for spectral deconvolution. Data were normalized to telmisartan (internal standard, *m/z* for [M + H]$^+$ 515.2447). Automatic processing detected 629 and 557 features for the *AbP450-5021* VIGS vs. control lines and *AbP450-116623* VIGS vs. control lines data sets, respectively. These data were exported to EZinfo v3.0.2 software (Umetrics, Umeå, Sweden) which filtered signals further to yield 486 and 440 variables for statistical analyses. For orthogonal projections to

latent structures discriminant analysis (OPLS-DA), Pareto scaling was implemented. Feature ions were selected for UHPLC/MS/MS analyses to support annotations based on the following criteria: p(corr) score below −0.3 and greater than +0.3, *m/z* ≥ 125 and even *m/z* values for [M + H]$^+$. These criteria were established empirically as they yielded metabolites with *P*-values < 0.02 in at least one VIGS genotype.

**Statistical analyses.** T-tests (unpaired, two-tailed, Welch correction) were conducted using GraphPad Prism 9.3.1 and Microsoft Excel 2013, and *P* values < 0.02 were considered statistically significant. False discovery rate *q*-values were calculated using Progenesis QI software.

**Computational phylogenetic analyses.** Protein sequences were aligned with ClustalW and neighbor-joining trees were constructed with MEGA 6.06[50]. The complete deletion option was used and bootstrap values were determined from 1000 replicates and evolutionary distances computed using the Poisson correction method.

**Chemical synthesis of acyl pseudotropine and acyl tropine.** For synthesis of acylated pseudotropine and acylated tropine derivatives with an acetyl, propionyl, isobutyryl or isovaleryl group, a mixture of 1 mmol pseudotropine or tropine free base in 10 ml chloroform, 1.1 mmol hydrogen chloride (4 M solution in dioxane) and 2 mmol acyl chloride was stirred overnight at 50 °C. The mixture was reduced to a volume of approximately 2 mL using a vacuum rotary evaporator and subsequently extracted with 5 mL water. The water phase was washed twice with 2.5 mL chloroform, adjusted to pH 13 with 10 M sodium hydroxide solution (in water) and extracted four times with either ethyl acetate or hexane. The collected ethyl acetate or hexane phases were combined, and the solvent was evaporated under reduced atmospheric pressure. For synthesis of tigloyl pseudotropine and tigloyl tropine, a mixture of 20 mmol pseudotropine or tropine free base in 200 mL chloroform, respectively, 22 mmol hydrogen chloride in dioxane (4 M solution) and 40 mmol acyl chloride was stirred overnight at 50 °C. The mixture was reduced to a volume of approximately 20 mL using a vacuum rotary evaporator and subsequently extracted with 50 mL water. The water phase was washed twice with 50 mL chloroform, then adjusted to pH 13 with 10 M sodium hydroxide solution (in water) and extracted four times with 20 mL hexane. The collected hexane phases were combined and dried with anhydrous sodium sulfate. Tigloyl pseudotropine or tigloyl tropine was precipitated from the extract through dropwise addition of 20 mmol hydrogen chloride (4 M solution in dioxane), and the hexane supernatant was removed by pipetting. The precipitate was washed six times with hexane and dried under reduced atmospheric pressure. The acylated pseudotropine and tropine derivatives were analyzed by UHPLC/MS/MS and NMR spectroscopy (Supplementary Data 3, Supplementary Data 6).

**Chemical synthesis of tigloyl norpseudotropine.** Two millimol norpseudotropine hydrochloride was dissolved in 4 mL trifluoroacetic acid by stirring at room temperature for 5 min. To this solution, 385 μL tigloyl chloride was added drop by drop. The mixture was stirred for 4 hours at room temperature and then cooled using an ice-water bath. Up to 90 mL diethyl ether was added drop by drop to give a fine white precipitate that was subsequently sedimented through centrifugation. The precipitated tigloyl norpseudotropine hydrochloride was washed six times with diethyl ether and dried under a stream of nitrogen gas. Tigloyl norpseudotropine was verified by UHPLC/MS/MS and NMR spectroscopy (Supplementary Data 3, Supplementary Data 6).

**NMR spectroscopy.** The target compounds dissolved in deuterated NMR solvent (D₂O or CDCl₃, Sigma-Aldrich) were transferred to an NMR tube for analysis (Supplementary Data 6). ¹H, ¹³C, COSY, HSQC, HMBC, ¹H-¹H *J*res, and NOESY NMR experiments were performed using Agilent DDR2 500 MHz NMR and Varian 600 MHz superconducting NMR spectrometers with VnmrJ 3.2. Spectra were referenced to non-deuterated chloroform solvent signals (δH = 7.26 (s) and δC = 77.2 (t) ppm). NMR spectra were processed using MestReNova 12.0.1 software (Mestrelab Research).

**Transient assays in *N. benthamiana* leaves.** The *N. benthamiana* plants were grown for 3.5 to 4 weeks in Sunshine® Redi-Earth Plug and Seedling Mix at 25 °C under a 14-h photoperiod at 150 μmol m⁻² s⁻¹ light intensity. The full-length open reading frames of *AbP450-5021* (MH292965) and *AbP450-116623* (MH292967) were inserted into pEAQ-HT-DEST1[51]. Transformants of *A. tumefaciens* LBA4404 carrying these binary vectors or the empty vector (control) were grown overnight at 30 °C in Luria-Bertani liquid medium containing 50 μg mL⁻¹ rifampicin and 50 μg mL⁻¹ kanamycin. The *A. tumefaciens* cells were sedimented by centrifugation at 3800 × *g* for 8 min, washed in infiltration buffer (10 mM MgCl₂ in 10 mM MES-KOH pH 5.7), and resuspended in infiltration buffer containing 200 μM acetosyringone. The optical density at OD₆₀₀ of the suspensions was adjusted to 0.8, and the suspensions were incubated for 30 min at 30 °C prior to infiltration into *N. benthamiana* leaves. To co-produce AbP450-5021 and AbP450-116623 in *N. benthamiana*, equal volumes of the two *A. tumefaciens* suspensions

were mixed and then infiltrated into *N. benthamiana* leaves using a needleless syringe. The plants were then incubated in the growth chamber under standard conditions. Four days post-infiltration, the infiltrated leaf areas were infiltrated with 1 mM substrate hydrochloride in water. For comparative analyses, only plants from the same batch were used. For each condition, a total of three leaves, each on a different plant from a different flat, were infiltrated as biological replicates. Six days after infiltration with *A. tumefaciens* suspension, 40 mg tissue was collected from each infiltrated leaf area in a 2-mL plastic tube containing three 3 mm glass beads and frozen in liquid nitrogen. The frozen leaf tissue was ground to fine powder using a TissueLyser II (Qiagen). For metabolite extraction, 400 μL extraction solvent (20% methanol, 0.1% formic acid and telmisartan in water) was added to each sample, and the samples were incubated for 3 h on an orbital shaker at room temperature in the dark. The samples were centrifuged at maximum speed for 15 min and supernatants were transferred into fresh vials. UHPLC/MS analyses were performed using a Waters Xevo G2-XS Q-TOF mass spectrometer equipped with a Shimadzu LC-20AD HPLC system. For analysis of extracts from assays with acylated substrates, the leaf extracts were diluted 1:1000 in 100 mM ammonium formate, 1% formic acid and 1% methanol in water to a final telmisartan (internal standard) concentration of 0.2 μM. A 10-μL volume of each extract was subsequently analyzed using a 10-min gradient elution method on an Ascentis Express F5 column (MilliporeSigma, pentafluorophenylpropyl stationary phase, 2.1 × 100 mm with 2.7 μm particle size) at 40 °C column temperature with mobile phases consisting of 100 mM ammonium formate and 1% formic acid in water (solvent A) and methanol (solvent B). The 10-min gradient employed 1% B at 0.00 to 1.00 min, a linear gradient to 50% B at 2.00 min, a linear gradient to 100% B at 8.00 min followed by a return to 1% B and a hold from 8.01 to 10.00 min. The flow rate was 0.3 mL/min. For analysis of extracts from assays with non-acylated substrates, leaf extracts were diluted 1:100 in acetonitrile to a final telmisartan concentration of 0.2 μM. A 10-μL volume of each extract was subsequently analyzed using a 10-min gradient elution method on a CORTECS HILIC column (silica-based stationary phase, 2.1 × 100 mm, 1.6 μm particle size) at 40 °C column temperature with mobile phases consisting of 100 mM ammonium formate and 1% formic acid in water (solvent A) and acetonitrile (solvent B). The 10-min HILIC method gradient was as follows: 95% B at 0.00 to 1.00 min, linear gradient to 40% B at 8.00 min followed by a return to 95% B and held from 8.01 to 10.00 min at a flow rate of 0.3 mL/min. The mass spectrometer was equipped with an electrospray ionization source and operated at 3000 V capillary voltage, 40 V cone voltage, 100 °C source temperature, 350 °C desolvation temperature, 20 L/h cone gas flow, and 500 L/h desolvation gas flow. Mass spectrum acquisition was performed in positive ion mode over *m/z* 50 to 1,000 with scan time of 0.25 s. Data were analyzed for exact *m/z* values corresponding to [M + H]⁺ positive ions of candidate substrates, putative *N*-demethylated and/or hydroxylated products with up to four hydroxy groups.

### Production and purification of tigloyl hydroxynorpseudotropine isomers from *N. benthamiana*. Four to five weeks old *N. benthamiana* plants were infiltrated with *A. tumefaciens* LBA4404 cells carrying pEAQ-HT-DEST1 constructs for expression of either *AbP450-5021* or *AbP450-116623* as described above. The treated plants were further incubated in the plant growth chamber under standard conditions. Four days post-infiltration, the *A. tumefaciens*-infiltrated leaves were infiltrated with 1 mM tigloyl norpseudotropine hydrochloride solution in water, and the plants were re-transferred to the growth chamber. After two days, the treated leaves were collected, frozen in liquid nitrogen, ground to fine powder with mortar and pestle, and the leaf powder was stored at −80 °C.

Tigloyl hydroxynorpseudotropine isomers 1 and 2 were purified each from an equivalent of 400–500 g *N. benthamiana* leaf fresh weight. Crude metabolite extracts were prepared from 100 g leaf tissue aliquots stirred in 700 mL 20% methanol and 0.1% formic acid in water for 3 h at room temperature in the dark. The suspension was then vacuum-filtered over a bed of celite. The pH of the filtered extract was adjusted to 6 with ammonium hydroxide solution (in water), and the solvent was removed under reduced atmospheric pressure. A series of solvent fractionation steps was used to remove some of the major contaminants including nicotine from the crude extract and enrich the target compound. The crude extract was reconstituted in 50 mL water, basified to pH 11 with ammonium hydroxide solution followed by five times liquid-liquid partitioning against 100 mL ethyl acetate. The ethyl acetate layers were collected, combined, washed with 0.2% ammonium hydroxide solution (in water), and the solvent was removed under reduced pressure in a rotary evaporator. Dried ethyl acetate extract (equivalent to 200 g leaf fresh weight) was reconstituted in 50 mL 0.2% ammonium hydroxide solution followed by five times liquid-liquid partitioning against 100 mL hexane. The hexane layers were collected, combined, washed with 0.2% ammonium hydroxide solution, and the solvent was removed under reduced pressure in a rotary evaporator. The dried hexane extract was reconstituted in 50 mL 0.2% ammonium hydroxide solution followed by five times liquid-liquid partitioning against 100 mL methyl *tert*-butyl ether. The methyl *tert*-butyl ether layers were collected, combined, washed with 0.2% ammonium hydroxide solution, and the solvent was removed under reduced pressure in a rotary evaporator. The extract was reconstituted in 2.2 mL 100 mM ammonium formate and 1% formic acid in water. Preparative HPLC fractionation was performed using a Waters 2795 HPLC system equipped with an Ascentis Express F5 column (pentafluorophenylpropyl

stationary phase, 4.6 × 150 mm, 2.7 μm particle size) and coupled to a LKB BROMMA 2211 fraction collector. The sample injection volume was 200 μL. The mobile phases consisted of 100 mM ammonium formate and 1% formic acid in water (solvent A) and 100 mM ammonium formate and 1% formic acid in acetonitrile/water 8:2, by volume (solvent B). A 30-min gradient elution method was used: 1% B at 0.00 to 1.00 min, a linear gradient to 8% B at 6.50 min, a linear gradient to 15% B at 20.00 min, a linear gradient to 100% B at 28.00 min followed by a return to 1% B and a hold from 28.01 to 30.00 min at a flow rate of 1 mL/min. Fractions were collected at 30-s intervals. The abundance of the target compounds in the collected fractions was confirmed by UHPLC/MS analyses using a Waters Xevo G2-XS Q-TOF mass spectrometer equipped with a Shimadzu LC-20AD HPLC system. Aliquots of the collected fractions were 1:1000 diluted in 100 mM ammonium formate, 1% formic acid and 1% methanol in water. A 10-μL volume of each sample was analyzed using a 10-min gradient elution method on an Ascentis Express F5 column (pentafluorophenylpropyl stationary phase, 2.1 × 100 mm with 2.7 μm particle size) at 40 °C column temperature with mobile phases consisting of 100 mM ammonium formate and 1% formic acid in water (solvent A) and methanol (solvent B). The 10-min gradient elution method was as follows: 1% B at 0.00 to 1.00 min, a linear gradient to 50% B at 2.00 min, a linear gradient to 100% B at 8.00 min followed by a return to 1% B and a hold from 8.01 to 10.00 min. The flow rate was 0.3 mL/min. Tigloyl hydroxynorpseudotropine isomers 1 and 2 were collected in fractions 17–23 and 15–20 from the F5 column, respectively. Selected fractions were kept for 15 min under a flow of nitrogen gas, basified to pH 11 with ammonium hydroxide and subjected to five times liquid-liquid partitioning against 2 mL methyl *tert*-butyl ether. The methyl *tert*-butyl ether layers were combined, and the solvent was evaporated under a stream of nitrogen gas. Samples were analyzed by NMR spectroscopy.

**Reporting summary**. Further information on research design is available in the Nature Research Reporting Summary linked to this article.

## Data availability
The source data underlying Fig. 2, Supplementary Data 1, and Supplementary Figs. 1–5 are provided as a Source Data file. NMR data including NMR spectra are provided as Supplementary Data 6. The following structures have been deposited in NCBI PubChem: 3-tigloyl (pseudo)tropine (CID 12444363), 3-tigloyl nor(pseudo)tropine (CID 91750091), tigloyl hydroxynorpseudotropine isomer 1 (CID 155907168), tigloyl hydroxynorpseudotropine isomer 2 (CID 155907169), 3-acetyl (pseudo)tropine (CID 10559369), 3-propionyl (pseudo)tropine (CID 91751182), 3-isobutyryl (pseudo)tropine (CID 3083598), 3-isovaleryl (pseudo)tropine (CID 156008899), and 3-senecioyl (pseudo)tropine (CID 13874122). Metabolomics data have been deposited to the EMBL-EBI MetaboLights database with the identifier MTBLS5041. Source data are provided with this paper.

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

## Acknowledgements

We thank Zack Wojcik at Michigan State University for technical assistance. NMR spectra were collected at the Max T. Rogers NMR Facility at Michigan State University. We thank Professor George Lomonossoff (John Innes Centre) for providing the pEAQ vectors. This research was supported by National Science Foundation's Plant Genome Research Program award IOS-1546617 to C.S.B. and A.D.J. M.A.B. was supported in part by an assistantship from the Plant Breeding, Genetics and Biotechnology Graduate Program at Michigan State University. C.S.B. and A.D.J. are supported in part by Michigan AgBioResearch and through USDA National Institute of Food and Agriculture, Hatch project numbers MICL02552 and MICL02474.

## Author contributions

R.S. and C.S.B. conceived the study, and R.S., A.D.J. and C.S.B. designed the research. C.S.B. and J.M.G. generated plant expression constructs. J.M.G. and M.A.B. contributed the set of *A. belladonna* AbP450-5021 VIGS/control lines, and R.S. and J.M.G. contributed the set of AbP450-116623 VIGS/control lines. J.M.G. performed qRT-PCR analyses and analyzed the data. J.M.G. and M.A.B. conducted initial UHPLC/MS analyses indicating the AbP450-5021 VIGS lines were impaired in calystegine biosynthesis. R.S. conducted the untargeted UHPLC/MS, multivariate statistical and UHPLC/MS/MS analyses. R.S. and A.D.J. annotated the metabolites. M.A.B. and R.S. synthesized acyl pseudotropine and acyl tropine candidate substrates, and R.S. developed the method for the synthesis of tigloyl norpseudotropine. R.S. conducted the transient biochemical assays in *N. benthamiana*, and analyzed and interpreted the data. R.S. and A.D.J. designed the method for the purification of tigloyl hydroxynorpseudotropine isomers 1 and 2 from up-scaled transient assays, and R.S. purified both isomers. T.A. conducted the NMR experiments and interpreted the NMR data. A.D.J. reviewed the NMR interpretation. R.S. performed the phylogenetic analyses. R.S. wrote the article, and C.S.B. and A.D.J. edited the manuscript.

## Competing interests

The authors declare no competing interests.
