## [Peer Review File · Nature Communications]

Metabolomics-guided discovery of cytochrome P450s involved in pseudotropine-dependent biosynthesis of modified tropane alkaloidsREVIEWER COMMENTS

Reviewer #1 (Remarks to the Author):

The article by Barry and coworkers describes the extensive metabolomic analysis and the subsequent identification of two CYP450 enzymes involved in tropane biosynthesis. The work is well done, with extensive metabolomics and products are well characterized. However, my personal opinion is that this work is better suited for a more specialized journal (e.g. Plant Physiology). The use of extensive metabolomics-derived metabolic networks to enhance gene discovery is not new, and the biochemical reactions catalyzed by these P450s (demethylation, hydroxylation) are well known.

Reviewer #2 (Remarks to the Author):

Sadre et al. describe identification of enzymatic activities of cytochrome P450s in alkaloid biosynthesis in *A. belladonna*. The authors generated virus-induced gene silenced *A. belladonna* lines and compared their metabolite profiles to control plants. This allowed identifying the biochemical functions of the CYPs without a priori knowledge. Transient overexpression of the genes in tobacco allowed performing in planta assays that confirmed the findings.

The topic is definitely interesting and worth publishing. The study is performed at a high technical standard.

However, the authors state several times that their metabolite profiling approach was untargeted. I do not agree to that. The authors used a number of settings that specifically alkaloid metabolites were detected (e.g. presence of nitrogen was a prerequisite). This does, of course not lower the value of the manuscript but I suggest removing the term "untargeted" since this was clearly not an untargeted study. Obviously, from the first moment on the authors wanted to investigate whether these CYPs have a function in alkaloid biosynthesis and designed the study according to that.

Although the study is important, a clear drawback is its length. The authors describe every detail in length, which makes it sometimes boring to read the manuscript. The discussion is very long (7 pages!) and large parts of it (particularly the first 3 pages) reiterate just the results that were anyway presented and discussed a few pages ahead. It would be good if the authors could make the manuscript more concise. There is, however, no need to shorten the Materials and Methods section; this part is very well written. Similarly, also the figures are fine.

From the introduction and the first paragraph of the results section it does not become clear why particularly those two CYPs were used as candidates. It would be good if the rationale using particularly those could be explained clearly.

Reviewer #3 (Remarks to the Author):

The manuscript submitted by Sadre et al reported the identification of uncharacterized P450s in *Atropa belladonna* using reverse genetic approaches including VIGS and Agrobacterium mediated transient gene expression. The experiments shown are elegant and well documented to support the N-demethylation and ring hydroxylase activities of CYP450-5021 (CYP71 family) and ring hydroxylase activity of AbP450-116623 (CYP82 family). Whereas this manuscript would be worth for the publication, some points should be declared to increase the importance of this manuscript.

1) Whereas authors mentioned that changes of some tropane alkaloids in root tissues were marginal after VIGS of two P450s, tropane alkaloids and calystegines are transported to aerial parts from root. So, characterization of metabolites in aerial parts would be needed to clarify the physiological effects

of these P450s in planta. In addition, quantitative data, i.e., absolute value of some key alkaloids, such as hyoscyamine, scopolamine, major calystegines would be useful to estimate the contribution of P450s in metabolic network.

2) Identification of N-demethylation activity of AbP450-5021 is quite important. Since conservation of certain amino acid residues of P450s for hydroxylase activity is well known, authors may discuss the unique amino acid residues of AbP450-5021 for novel activity.

Minor comments;

Title "Metabolomics-guided discovery " might be modified since reverse genetics would be crucial to identify P450 functions; functional identification of P450s with N-demethylation would be more adequate.

Responses to Reviewer Critiques (author responses in blue)

Reviewer #1 (Remarks to the Author):

The article by Barry and coworkers describes the extensive metabolomic analysis and the subsequent identification of two CYP450 enzymes involved in tropane biosynthesis. The work is well done, with extensive metabolomics and products are well characterized. However, my personal opinion is that this work is better suited for a more specialized journal (e.g. Plant Physiology). The use of extensive metabolomics-derived metabolic networks to enhance gene discovery is not new, and the biochemical reactions catalyzed by these P450s (demethylation, hydroxylation) are well known.

Au: We are glad reviewer #1 appreciates that the work is well done. In response to the comment that metabolomics-explored networks for gene discovery is not a new concept and that cytochrome P450-catalyzed reactions are well known, we respond with a reminder that the objective of this investigation was not to establish novel reaction types, but highlight how transcript-guided metabolomics combined with *in planta* assays leads to **discoveries** of key intermediates and enzyme functions (i.e. substrate recognition) and addresses well-known bottlenecks and challenges that have limited progress in engineering specialized metabolism in plants.

Reviewer #2 (Remarks to the Author):

Sadre et al. describe identification of enzymatic activities of cytochrome P450s in alkaloid biosynthesis in *A. belladonna*. The authors generated virus-induced gene silenced *A. belladonna* lines and compared their metabolite profiles to control plants. This allowed identifying the biochemical functions of the CYPs without a priori knowledge. Transient overexpression of the genes in tobacco allowed performing *in planta* assays that confirmed the findings.

The topic is definitely interesting and worth publishing. The study is performed at a high technical standard.

However, the authors state several times that their metabolite profiling approach was untargeted. I do not agree to that. The authors used a number of settings that specifically alkaloid metabolites were detected (e.g. presence of nitrogen was a prerequisite). This does, of course not lower the value of the manuscript but I suggest removing the term “untargeted” since this was clearly not an untargeted study. Obviously, from the first moment on the authors wanted to investigate whether these CYPs have a function in alkaloid biosynthesis and designed the study according to that.

Au.: We are grateful that reviewer #2 appreciates that this study represents a high technical standard. With regard to whether this study invokes an untargeted approach to metabolomics, we propose that “untargeted” is a term that has been used extensively in the literature to describe metabolomics studies. A 2016 review (DOI: 10.1007/s13361-016-1469-y) mentioned that a “... major advantage of untargeted metabolomics is the collection of data without preexisting knowledge; however, this is accompanied with the caveat that certainly sample

preparation and analytical methods have a direct impact on the qualitative results that are obtained.” In our study, the analytical protocols employed were designed to capture a wide range of metabolites not limited to alkaloids without foreknowledge of the identities of key metabolic intermediates. All metabolomics studies, including ours, involve some filtering of data for a variety of good reasons, and we acknowledge that our data processing removed some metabolites unrelated to potential (and often unknown) alkaloid pathways to minimize type I errors that arise from a larger number of measurements than samples, as well as to facilitate mechanistic discussion of results. In contrast to targeted metabolite profiling, many of the alkaloids discovered in this study have not been previously reported in *Atropa belladonna* or any other species in the Solanaceae family. These compounds would have remained unnoticed in an entirely targeted analysis. In deference to the reviewer, we have replaced the term “untargeted metabolomics” in the manuscript with “discovery metabolomics” to describe the entirety of the approach, reflecting that *a priori* knowledge of metabolite identities was not available, while retaining the term “untargeted” to describe the UHPLC/HRMS analytical procedures which cast as broad a net for metabolite capture in the raw data as our extraction and analytical protocols allow (prior to data filtering).

Although the study is important, a clear drawback is its length. The authors describe every detail in length, which makes it sometimes boring to read the manuscript. The discussion is very long (7 pages!) and large parts of it (particularly the first 3 pages) reiterate just the results that were anyway presented and discussed a few pages ahead. It would be good if the authors could make the manuscript more concise. There is, however, no need to shorten the Materials and Methods section; this part is very well written. Similarly, also the figures are fine.

Au.: The text was revised to address the reviewer’s comment, remove redundancy, and improve clarity. The discussion section is now reduced to four pages in length, and we are convinced that the manuscript is improved without sacrificing our goal of explaining key aspects of the study to help readers follow the reasoning underlying the experiments.

From the introduction and the first paragraph of the results section it does not become clear why particularly those two CYPs were used as candidates. It would be good if the rationale using particularly those could be explained clearly.

Au.: As stated in the introduction of the manuscript (lines 63-67), these CYPs were initially selected as candidates in an earlier study we performed that identified CYP82M3 as tropinone synthase (*Bedewitz MA, Jones AD, D’Auria JC, Barry CS. Tropinone synthesis via an atypical polyketide synthase and P450-mediated cyclization. Nat Commun. (2018) doi: 10.1038/s41467-018-07671-3*). In that study, we selected CYPs based solely on their preferential expression in roots compared with other tissues, a well-established strategy for selecting candidate genes involved in tropane alkaloid biosynthesis. This study initially included AbP450-5021 and AbP450-116623, but the encoded enzymes did not possess tropinone synthase activity and the genes remained functionally uncharacterized.

Our preliminary targeted metabolite analysis on root tissue from *Atropa* VIGS lines suggested that these CYPs may be involved in calystegine biosynthesis. In the current study, we repeated the VIGS experiments with a larger number of biological replicates and combined VIGS with a discovery metabolomics-based approach to determine the functions of these CYPs. We have added brief explanation in the Introduction regarding how we selected these candidate genes.

Reviewer #3 (Remarks to the Author):

The manuscript submitted by Sadre et al reported the identification of uncharacterized P450s in *Atropa belladonna* using reverse genetic approaches including VIGS and *Agrobacterium* mediated transient gene expression. The experiments shown are elegant and well documented to support the N-demethylation and ring hydroxylase activities of CYP450-5021 (CYP71 family) and ring hydroxylase activity of AbP450-116623 (CYP82 family). Whereas this manuscript would be worth for the publication, some points should be declared to increase the importance of this manuscript.

1) Whereas authors mentioned that changes of some tropane alkaloids in root tissues were marginal after VIGS of two P450s, tropane alkaloids and calystegines are transported to aerial parts from root. So, characterization of metabolites in aerial parts would be needed to clarify the physiological effects of these P450s in planta. In addition, quantitative data, i.e., absolute value of some key alkaloids, such as hyoscyamine, scopolamine, major calystegines would be useful to estimate the contribution of P450s in metabolic network.

Au.: We appreciate that the reviewer found our experiments to be elegant and well-documented. The reviewer is correct in that tropane alkaloids are transported from the roots to the shoots in members of the Solanaceae. However, while transport of tropanes is an interesting phenomenon, it is not well understood and it is unclear which specific alkaloids are transported. As tropanes are synthesized in the roots in *Atropa belladonna*, and our study is focused on biosynthesis and pathway discovery, not transport, we directed our metabolite profiling strategy solely on the roots. Aerial tissue samples from these experiments were not harvested because the genes we silenced are under normal conditions root-preferentially expressed and either not expressed in aerial tissues (*AbP450-116623*) or expressed at a very low level (*AbP450-5021*) (Supplementary Figure 1). Following the discovery of a number of novel alkaloids in this study, transport could represent topics for future work but it is beyond the scope of the current study. Even if we detected the presence of some of these novel tropanes in aerial tissues, it would not provide any information on the physiological effects of these metabolites, as suggested by the reviewer, as this would require further work to investigate.

Although we appreciate the value in absolute quantitative measurements, much time has passed since the extracts were analyzed and we did not generate calibration curves for absolute concentrations of these alkaloids at the time nor assess matrix effects that might influence calculations of absolute levels. Since mass spectrometer instrument responses drift with time, (and the project was not focused on constructing a quantitative network for all tropane alkaloid metabolism, but rather to establish metabolic connections) we are not

convinced that generating new calibration curves at this point would make a significant contribution to our findings.

2) Identification of N-demethylation activity of AbP450-5021 is quite important. Since conservation of certain amino acid residues of P450s for hydroxylase activity is well known, authors may discuss the unique amino acid residues of AbP450-5021 for novel activity.

Au.: We appreciate that the reviewer recognizes that the *N*-demethylation activity of AbP450-5021 represents an important discovery. Please note that during oxidative *N*-demethylation, a cytochrome P450 catalyzes a hydroxylation reaction that precedes spontaneous conversion of the hydroxylated intermediate in the *N*-demethylated product and the by-product formaldehyde. Thus, AbP450-5021's dual role as *N*-demethylase/ring-hydroxylase in modified tropane alkaloid biosynthesis is based on hydroxylase activity. Other hydroxylases of the CYP71 subfamily identified in previous studies may have *N*-demethylase activity that remained unrecognized. Since AbP450-5021 is the only known plant *N*-demethylase/ring-hydroxylase in the CYP71 subfamily, a comprehensive structural enzymology study on AbP450-5021 and related cytochrome P450s would be needed to assign residues for a specific catalytic activity. This is beyond the scope of the present study.

Minor comments;

Title "Metabolomics-guided discovery " might be modified since reverse genetics would be crucial to identify P450 functions; functional identification of P450s with *N*-demethylation would be more adequate.

Au.: We prefer to not change the title of the manuscript to one that focuses solely on *N*-demethylation of tropanes because as described above, *N*-demethylation is just one of several components of this study. The combination of gene silencing, transient *in planta* assays, metabolomics, and metabolite structure elucidation provides solid evidence regarding enzymes' native substrate promiscuity and is broader in scope than a focus on *N*-demethylation. As this approach is also applicable to other organisms, we believe the original title will connect with a broader audience.

REVIEWERS' COMMENTS

Reviewer #3 (Remarks to the Author):

In response letter, Sadre et al. explained the current situation of research conditions and some difficulties to continue the analyses of metabolites in aerial parts as well as more quantitative analyses of tropane alkaloids. Whereas it is pity that authors could not provide more information about these alkaloids in whole plants, the present manuscript would be still worth for the publication to understand the novel enzyme activities of Atropa P450s involved in the pseudotropine-derived alkaloids. In addition, the revised title would be more suitable for this manuscript. This revised manuscript would be ready for the publication.

Authors' response to Reviewer #3 remarks:

In response letter, Sadre et al. explained the current situation of research conditions and some difficulties to continue the analyses of metabolites in aerial parts as well as more quantitative analyses of tropane alkaloids. Whereas it is pity that authors could not provide more information about these alkaloids in whole plants, the present manuscript would be still worth for the publication to understand the novel enzyme activities of Atropa P450s involved in the pseudotropine-derived alkaloids. In addition, the revised title would be more suitable for this manuscript. This revised manuscript would be ready for the publication.

Au.: We are grateful for the reviewer's support in favor of this manuscript. With regard to the article's title, we would like to bring to your attention that this study heavily relied on untargeted metabolomics combined with multivariate statistics discovering many new alkaloids and the functions of the two cytochrome P450s in a previously largely unknown alkaloid subnetwork. Our article reports *N*-methylated alkaloids, *N*-demethylated alkaloids and links to other alkaloids (see Figs. 2 and Supplementary Data 1). Therefore, we would like to keep the original title with the emphasis on metabolomics. We are convinced that this metabolomics-guided approach is applicable to other systems to address key challenges in the discovery of metabolic subnetworks and novel genes for biotechnological applications.